# Pathway-divergent coupling of 1,3-enynes with acrylates through cascade cobalt catalysis

Heng Wang[1], Xiaofeng Jie[1], Qinglei Chong[1] ✉ & Fanke Meng [1,2,3,4] ✉

Catalytic cascade transformations of simple starting materials into highly functionalized molecules bearing a stereochemically defined multisubstituted alkene, which are important in medicinal chemistry, natural product synthesis, and material science, are in high demand for organic synthesis. The development of multiple reaction pathways accurately controlled by catalysts derived from different ligands is a critical goal in the field of catalysis. Here we report a cobalt-catalyzed strategy for the direct coupling of inexpensive 1,3-enynes with two molecules of acrylates to construct a high diversity of functionalized 1,3-dienes containing a trisubstituted or tetrasubstituted olefin. Such cascade reactions can proceed through three different pathways initiated by oxidative cyclization to achieve multiple bond formation in high chemo-, regio- and stereoselectivity precisely controlled by ligands, providing a platform for the development of tandem carbon-carbon bond-forming reactions.

Development of highly efficient and stereoselective cascade reactions to prepare important compounds in medicinal chemistry, total synthesis, and material science constitutes one of the most important areas in organic synthesis, as the benefits of cascade reactions include atom and step economy, as well as economies of time, labor, and waste generation[1–4]. Particularly, the rapid establishment of complex multifunctional molecules directly from readily available precursors and feedstocks evolves the way of organic synthesis. Alkynes are a class of easily accessible and versatile compounds. Catalytic regio- and enantioselective cascade double functionalization of alkynes provides a facile strategy for fast installation of multiple functional groups (Fig. 1a)[5]. However, only silyl, boryl, amino, and carboxy groups can be introduced[6–11]. Enantioselective intermolecular incorporation of multiple other carbon-carbon bonds remained unknown. Oxidative cyclization promoted by low-valent transition metal complexes represents a classical elementary step in organometallic chemistry. Catalytic regio- and enantioselective coupling of alkynes with aldehydes, imines, or alkenes through oxidative cyclization

promoted by Ni- and Co-based catalysts have received increasing interests[12–20]. Such reactions enabled direct stereoselective construction of functionalized compounds containing a multisubstituted alkene from simple starting materials (Fig. 1b). However, only one π bond of the alkyne and one molecule of aldehyde, imine, or alkene were able to participate in the coupling reactions through single oxidative cyclization. The resulting products were reluctant to undergo a second oxidative cyclization. As alkynes are preferential to undergo oxidative cyclization, it is difficult to incorporate two molecules of electrophiles into alkynes through oxidative cyclization. The only breakthrough was disclosed for Co-catalyzed tandem coupling of 1,3-enynes and ethylene to generate chiral cyclobutanes[21]. Two molecules of ethylene could be introduced through sequential double oxidative cyclization albeit with significant limitation of substrate scope. In addition, only a single reaction mode was revealed. To date, cascade regio- and stereoselective transformations of 1,3-enynes with two molecules of substituted alkenes through diverse pathways remained undisclosed.

[1]State Key Laboratory of Organometallic Chemistry, Center for Excellence in Molecular Synthesis, Shanghai Institute of Organic Chemistry, University of Chinese Academy of Sciences, 345 Lingling Road, Shanghai 200032, China. [2]State Key Laboratory of Elemento-Organic Chemistry, Nankai University, Tianjin 300074, China. [3]School of Chemistry and Materials Science, Hangzhou Institute for Advanced Study, University of Chinese Academy of Sciences, 1 Sub-lane Xiangshan, Hangzhou 310024, China. [4]Beijing National Laboratory for Molecular Sciences, Beijing 100086, China. ✉e-mail: chongql@sioc.ac.cn; mengf@sioc.ac.cn

**Fig. 1 | Background and reaction design. a** Catalytic enantioselective double functionalization of alkynes by sequential reactions. **b** Catalytic coupling of alkynes and activated alkenes through oxidative cyclization promoted by low-valent Ni or Co complexes. **c** Representative target molecules bearing multisubstituted alkenes. **d** Editing the carbon skeleton of the alkyne through a streamlined approach by introducing multiple C–C bonds. **e** Our cascade strategy for pathway-divergent coupling of 1,3-enynes with two molecules of acrylates to generate multifunctional 1,3-dienes.

Acrylates are feedstocks that are produced over three million tons annually. In this context, catalytic cascade reactions of alkynes and acrylates to construct a stereochemically defined trisubstituted or tetrasubstituted olefin with the simultaneous introduction of a stereogenic center and/or other functional groups is an attractive strategy, as a catalytic stereoselective synthesis of multisubstituted alkenes are nontrivial and such moieties widely exist in biologically active molecules (Fig. 1c). We envisioned that a streamlined approach for direct incorporation of two molecules of acrylates into the alkynes through oxidative cyclization followed by different elementary processes would enable divergent editing of the carbon skeleton of the alkyne, converting the alkyne to an alkene associated with the formation of two carbon-carbon single bonds or an alkane with two newly generated carbon-carbon double bonds or single bonds (Fig. 1d). Several factors impeded the development of such processes. Firstly, a multifunctional catalyst has to accurately control the chemo-, regio- and stereoselectivity of each step. It is challenging to introduce two molecules of electrophilic acrylates into the alkynes, as alkynes are intrinsically more prone to undergo oxidative cyclization with themselves and insertion into the metallacycle intermediates compared with acrylates due to its higher affinity to the metal center and lower-lying LUMO. For instance, examples of metal-catalyzed coupling of 1,6-diynes[22–25] and alkenes and 1:2 coupling of unsaturated hydrocarbons with carbonyls have been reported[26,27]. Moreover, for the combination of 1,3-enynes and acrylates, the regioselectivity of oxidative cyclization (four possibilities) and the selectivities for subsequent transformations are difficult to tune. Secondly, in comparison with the previous report on tandem reactions of 1,3-enynes with ethylene, much more possible reaction pathways of the metallocycle intermediates could complicate the transformations with acrylates. It is nontrivial to

| entry | ligand | yield (%)[a] | er[b] | entry | ligand | yield (%)[a] | er[b] |
|-------|--------|----------|-------|-------|--------|----------|-------|
| 1 | 3a | <5 | NA[c] | 8 | 3h | 80 | 88:12 |
| 2 | 3b | <5 | NA[c] | 9 | 3i | 74 | 4:96 |
| 3 | 3c | <5 | NA[c] | 10 | 3j | 89 | 5:95 |
| 4 | 3d | 87 | 93:7 | 11[d] | 3e | 90 | 95:5 |
| 5 | 3e | 87 | 95:5 | 12[e] | 3e | 93 | 94:6 |
| 6 | 3f | 85 | 6:94 | 13[f] | 3e | 92 | 94:6 |
| 7 | 3g | <5 | NA[c] | | | | |

**Fig. 2 | Optimization of reaction conditions.** [a]Yield of isolated product. [b]Enantiomeric ratios (er) were determined by analysis of HPLC spectra. [c]Not available. [d]The reaction was performed in DMA. [e]Mn was used as reductant. [f]The reaction was performed in the presence of 10 mol % CoI₂ and 10 mol % 3e with Mn in DMA. NMP = N-methylpyrrolidone, DMA = N,N-dimethylacetamide.

achieve multiple reaction modes. We considered that the development of a distinct reactivity paradigm for oxidative cyclization and non-traditional subsequent pathways for the metallocycle intermediates as well as proper choices of ligands and a metal center would address these longstanding critical issues.

Cobalt is an inexpensive earth-abundant transition metal of low toxicity[28]. Development of cobalt-catalyzed transformations to produce useful building blocks through the unique reactivity of organocobalt complexes fulfills the increasing demands for sustainable chemistry. Recently, we have successfully developed a cobalt-catalyzed protocol for enantioselective coupling of 1,1-disubstituted allenes and aldehydes through oxidative cyclization, generating three different sets of products from the same starting materials[29]. We reasoned that the combination of reaction design and judicious choices of chiral ligands enabled pathway-divergent oxidative cyclization and subsequent transformations[30–33]. We imagined that different ligands might be able to control the reaction pathway of the metallocycle intermediates formed from oxidative cyclization of 1,3-enynes and acrylates (Fig. 1e)[34–36]. We speculated that β-H elimination and protonation of the metallocycle would afford a highly reactive triene intermediate containing two 1,3-diene systems that might undergo a second oxidative cyclization with acrylates. A cobalt-based catalyst was expected to selectively react with one, furnishing highly functionalized products bearing a trisubstituted alkene and a stereogenic center (pathway a). In addition, direct insertion of the acrylates to the C−Co bond of the metallocycle would provide an approach for stereoselective formation of tetrasubstituted olefins (pathway b). Furthermore, trapping the intermediate generated from β-H elimination of the metallocycle containing a primary alkyl (R¹ = primary alkyl) with acrylates would trigger a series of cascade processes, delivering multifunctional building blocks (pathway c). In each pathway, competitive second oxidative cyclization with 1,3-enynes and insertion with 1,3-enynes have to be suppressed and chemoselective reactions with a second molecule of acrylate have to be precisely controlled by the catalysts. Here we report pathway-divergent coupling of readily available 1,3-enynes and acrylates by cascade cobalt catalysis to furnish highly functionalized

1,3-dienes bearing multisubstituted alkenes, accurately editing the carbon skeleton by constructing multiple carbon-carbon bonds.

## Results and discussion
### Reaction optimization
Our studies commenced with reaction of 1,3-enyne **1a** containing a tertiary alkyl group with benzyl acrylate **2a** in the presence of Co complexes derived from various chiral phosphine ligands (Fig. 2). Although no reaction occurred with the Co complexes derived from phosphinooxazoline **3a** (entry 1), bisphosphines bearing axial stereogenicity **3b** (entry 2), ferrocene skeleton **3c** (entry 3), and chiral phospholane fragments **3g** (entry 7), several phosphine ligands (**3d–f, 3h–j**) induced high enantioselectivity (entries 4–6, 8–10). Only product **4a** that incorporate two molecules of benzyl acrylate **2a** was afforded, and a single stereoisomer of the trisubstituted alkene was obtained. Performing the transformation in DMA led to a slight improvement of efficiency (entry 11). Screening of reductants revealed that Mn was equally effective (entry 12). Lowering the catalyst loading didn't reduce the efficiency (entry 13).

### Scope for enantioselective cascade coupling
With the optimal conditions, we investigated the scope of enantioselective cascade coupling of 1,3-enynes and acrylates (Fig. 3a). 1,3-Enynes bearing protected tertiary alcohol (**4b**) and quaternary centers substituted with three alkyls (**4c–d**) or two alkyls and one aryl group (**4e–w**) underwent enantioselective cascade coupling, producing 1,3-dienes in high efficiency and enantioselectivity (47–92% yield, 92:8–99:1 er). Electron-rich (**4g–h, 4m–n**), electron-deficient (**4f, 4i–k, 4q**), halogenated (**4k, 4q**), and sterically congested (**4m–q**) aryl groups are compatible with the reaction. 1,3-Enynes containing pharmaceutically important heterocycles such as thiophene (**4r–s**), benzofuran (**4t**), indole (**4u**), quinoline (**4 v**) and isoquinoline (**4w**) served as suitable substrates that afforded products in 60–83% yield and 94:6–99:1 er. The reaction of 1,3-enyne bearing ketal generated **4x** in 50% yield and 95.5:4.5 er. 1,3-Enynes substituted with secondary and primary alkyl groups were transformed to the coupling products in 43–70% yield and 94:6–96:4 er (**4y–z, 4aa–ac**). It is worth mentioning that

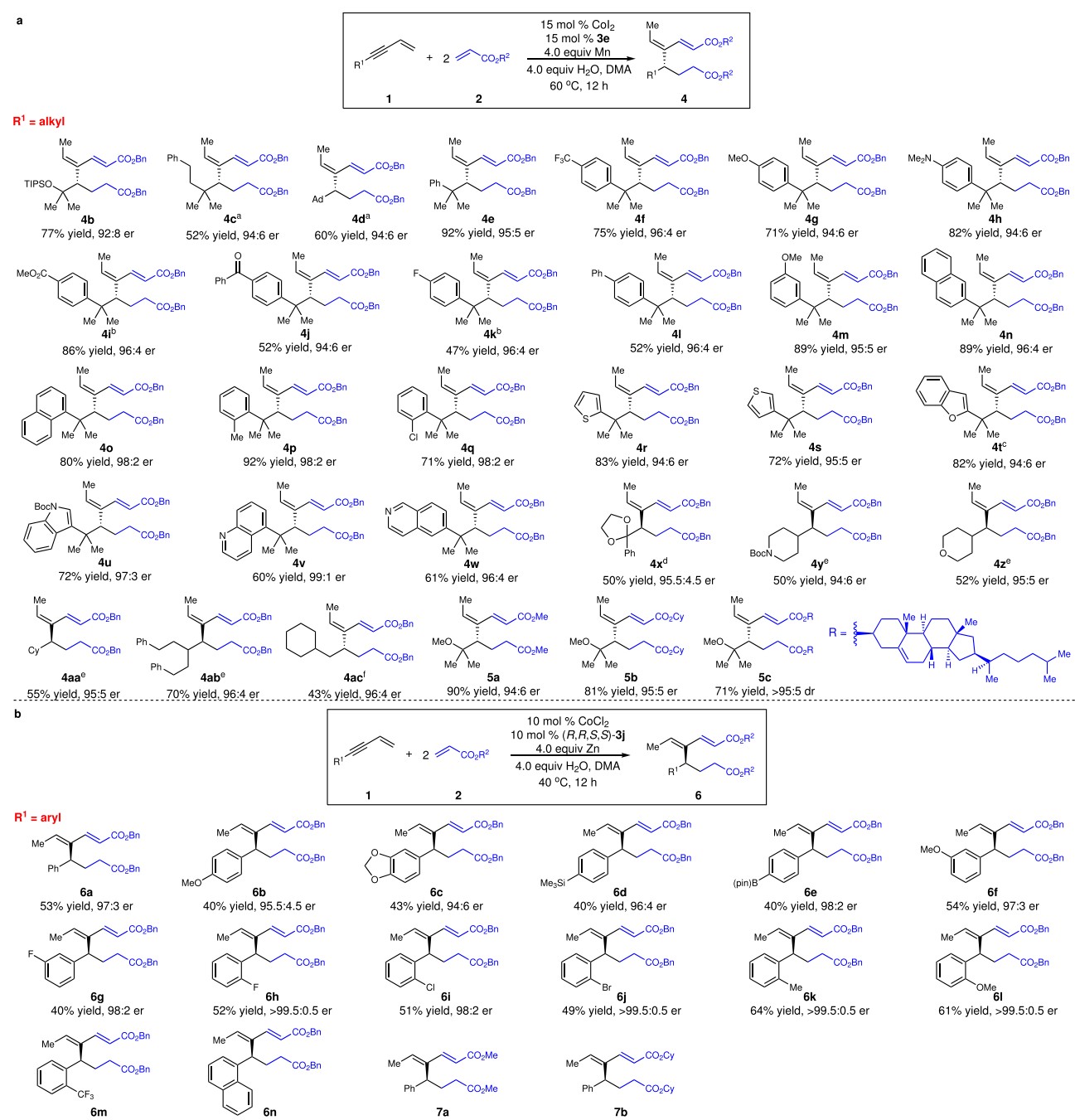

**Fig. 3 | Scope for Co-catalyzed enantioselective coupling of 1,3-enynes and acrylates. a** Scope for reactions with alkyl-substituted 1,3-enynes. **b**, Scope for reactions with aryl-substituted 1,3-enynes. [a]The reactions were performed with 10 mol % CoI₂, 10 mol % **3d** and Zn. [b]10 mol % catalyst loading. [c]20 mol % catalyst loading. [d]The reaction was conducted with 20 mol % CoI₂, 20 mol % (R,R,S,S)−**3j** and Zn. [e]The reactions were performed with 10 mol % CoI₂, 10 mol % **3f** and Zn at 50 °C. [f]The reaction was conducted with 10 mol % CoI₂, 10 mol % **3d** and Zn at 25 °C. DMA=N,N-dimethylacetamide, Boc -butyloxycarbonyl, pin pinacolato.

that a different chiral ligand was required for some substrates to achieve optimal efficiency and enantioselectivity (**4c−d**, **4x−z**, **4aa−ac**) (supplementary materials). Acrylates derived from other alcohols (**5a**, **b**), even complex natural product such as cholesterol (**5c**), are competent substrates.

We next surveyed the scope of aryl-substituted 1,3-enynes (Fig. 3b). Phosphine ligand **3j** provided the highest efficiency and enantioselectivity for reactions of aryl-substituted 1,3-enynes (**6a**, 53% yield, 97:3 er) (supplementary materials). A variety of 1,3-enynes bearing electron-deficient (**6g−h**, **6m**), electron-rich (**6b−c**, **6f**, **6l**), halogenated (**6g−j**), and sterically congested (**6h−n**) aryl groups could be converted into the

desired 1,3-dienes in 40−64% yield and 94:6− > 99.5:0.5 er. Functional groups such as silyl (**6d**) and boryl (**6e**) are compatible with the reaction. Acrylates derived from primary (**7a**) and secondary (**7b**) alcohols served as competent substrates. In all cases, a single isomer of the trisubstituted olefin was obtained. However, it is unexpected that the geometry of the trisubstituted alkene is different from that of products formed from reactions with alkyl-substituted 1,3-enynes.

## Cascade coupling to generate tetrasubstituted alkenes

Tetrasubstituted alkenes containing four carbon-based substituents are not only common units in biologically active molecules, but also

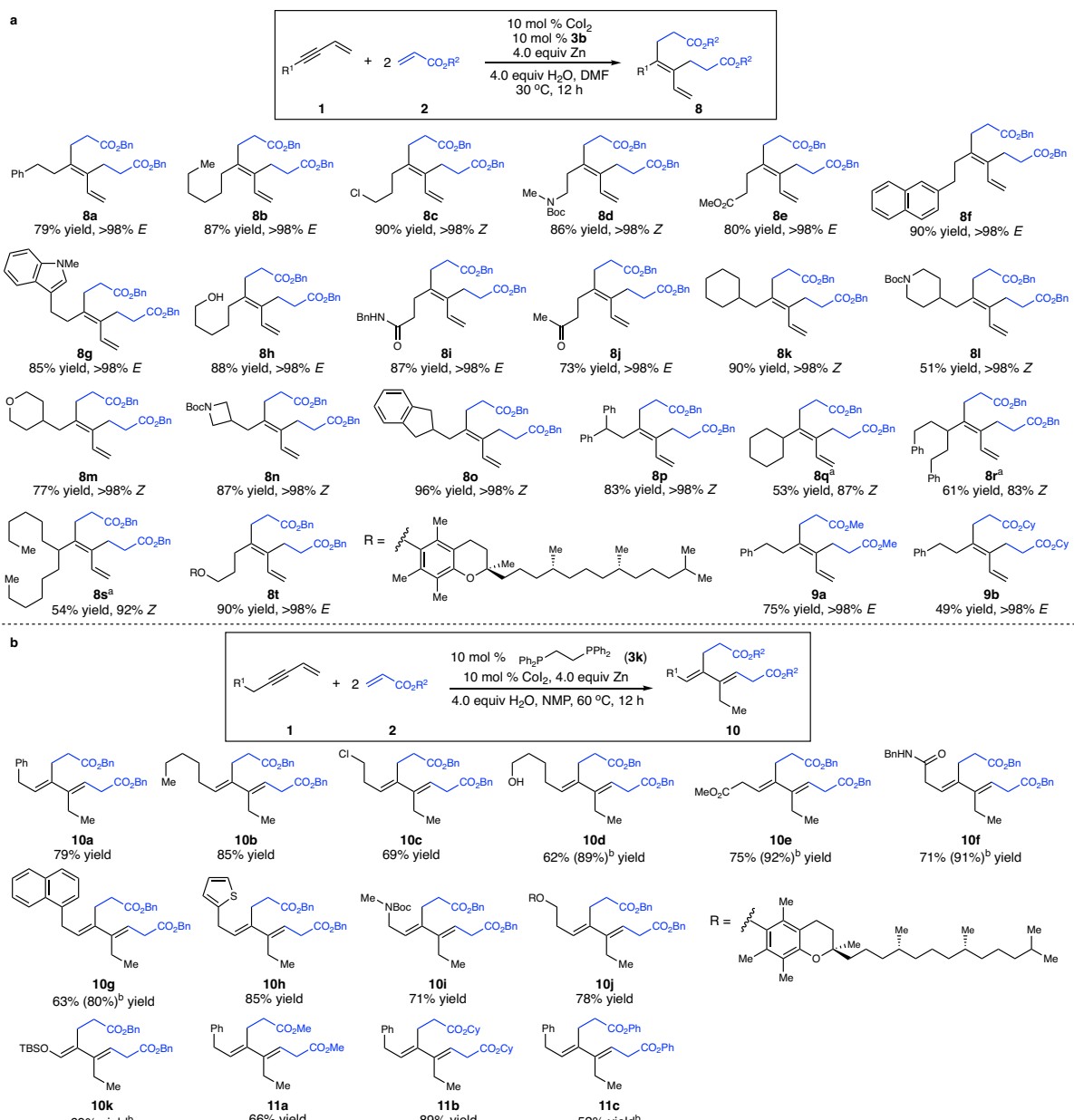

**Fig. 4 | Scope for Co-catalyzed coupling of 1,3-enynes and acrylates to generate achiral 1,3-dienes. a** Scope for Co-catalyzed coupling of 1,3-enynes and acrylates to afford 1,3-dienes bearing a tetrasubstituted alkene. **b** Scope for Co-catalyzed coupling of 1,3-enynes and acrylates to form achiral 1,3-dienes containing two trisubstituted olefins. [a]10 mol % **3c** was used. [b]20 mol % catalyst loading. NMP = N-methylpyrrolidone, DMF = N,N-dimethylformamide, Boc = t-butyloxycarbonyl, TBS = t-butyldimethylsilyl.

important intermediates that are versatile platforms to introduce a variety of functional groups into the carbon skeleton. Although catalytic double functionalization of internal alkynes constitutes a direct strategy for the introduction of two carbon-based substituents, significant limitations remained unsolved[37]. In particular, it is challenging to achieve simultaneous reactions of two molecules of alkenes with alkynes in the presence of a proton source, as protonation of the carbon–metal bond is competitive to the carbon-carbon bond forming reaction. Moreover, it is unknown the application of such strategy to the transformation of 1,3-enynes.

To this end, we found that reactions of 1,3-enynes bearing less sterically hindered primary and secondary alkyl groups in the presence of bisphosphine **3b** afforded *cis*-double alkylation products selectively (**8a–t**, Fig. 4a) (supplementary materials). Functional groups such as halogen (**8c**), amide (**8d, 8i**), ester (**8e**), free alcohol (**8h**), and ketone (**8j**) are well compatible with the reaction conditions. Substrates with

medicinally important heterocycles (**8g, 8l–n**) effectively underwent the cascade reaction, furnishing the 1,3-dienes containing a tetrasubstituted alkene as a single stereoisomer. Although 1,3-enynes substituted with secondary alkyl groups could participate in the reaction, isomerization of the tetrasubstituted alkene was observed (**8q–s**). Tertiary alkyl- and aryl-substituted 1,3-enynes are not reactive in the transformation, probably because larger steric hindrance of the substituent on 1,3-enyne retarded the insertion of the acrylate into the metallocycle intermediate. Functionalization of 1,3-enyne modified from complex natural product α-tocopherol furnished the 1,3-diene **8t** in 90% yield. Acrylates derived from primary (**9a**) and secondary (**9b**) alcohols are suitable substrates.

## Cascade coupling to generate products bearing two trisubstituted alkenes

Editing the carbon skeleton of alkynes by direct introduction of two carbon–carbon bonds constitutes a powerful strategy for the

construction of complex molecules. Few examples of saturating the triple bond by formation of two carbon-carbon σ-bonds have been reported, whereas catalytic approaches for connecting the two carbons of the alkyne with two π-bonds from two molecules of alkenes remained unsolved. Although enyne metathesis enabled saturation of the alkyne by generation of two π-bonds with one molecule of alkene, protocols for cascade reaction of two molecules of alkene to establish two trisubstituted olefins stereoselectively are in high demand[38].

We envisioned that tuning relative rates of β-H elimination, protonation and insertion with acrylate by a proper choice of ligand would provide a cascade pathway that might achieve this goal. We found that the reaction of primary alkyl-substituted 1,3-enyne **1b** with benzyl acrylate **2a** in the presence of Co complex derived from **3k** furnished the desired product **10a** in 79% yield (Fig. 4b) (supplementary materials). A wide range of functional groups such as halogen (**10c**), free alcohol (**10d**), ester (**10e**), and amide (**10f, 10i**) are well compatible. 1,3-Enynes bearing an aryl (**10g**) and heteroaryl (**10h**) were converted into the dienes in high efficiency as a single stereoisomer. Modification of complex natural product α-tocopherol afforded **10j** in 78% yield. Enol ether **10k** was formed as well through this protocol. Acrylates derived from methanol (**11a**), cyclohexanol (**11b**), and phenol (**11c**) served as suitable substrates. It is worth mentioning that only one isomer of the two trisubstituted alkenes was obtained. We also tested the three reaction pathways mentioned above for vinyl ethyl ketone, providing a complex mixture of unidentified products. Acrylamides and acrylonitrile were not reactive under three different reaction conditions above. In addition, 1,3-enynes bearing a 1,2- or 1,1-disubstituted alkene were not able to participate in the reactions for three different pathways.

## Scalable reactions and functionalization

The reactions can be performed on gram scale, affording **4a** in 77% yield with 93:7 er and **8a** in 90% yield (Fig. 5a). The multifunctional products generated from cascade coupling of 1,3-enynes and acrylates can be transformed into a variety of useful building blocks that are otherwise difficult to access. Rh-catalyzed diastereoselective 1,4-arylation of chiral diene **4a** with phenyl boronic acid provided **12** in 89% yield with 87.5:12.5 dr and **13** in 95% yield with 90:10 dr (Fig. 5b)[39]. The

stereochemistry of the products was solely controlled by the chiral ligands. Treatment of **4a** with $B_2(pin)_2$ and MeOH in the presence of NHC−Cu complex **14** followed by oxidative work-up furnished **15** as a single diastereomer (Fig. 5b)[40]. Diastereoselective 1,4-arylation of **6a** induced by chiral Rh complex derived from (S,S)−**3e** produced **16** in 81% yield and 87.5:12.5 dr associated with 9% 1,6-arylation product (Fig. 5c). We attempted developing a method for chemoselective transformation of one of the two ester groups in the product. Treatment of **6a** with $PhSO_3H$ in t-BuOMe gave **17** in 65% yield without erosion of enantioselectivity (Fig. 5c). Chemoselective hydrolysis of the methyl ester moiety with subsequent Curtius rearrangement afforded unnatural amino acid derivative **18** in 53% overall yield and 97:3 er [41]. Selective epoxidation of the tetrasubstituted alkene in diene **8a** delivered **19** as a single diastereomer (Fig. 5d)[42]. Ru-catalyzed E-selective cross metathesis with the terminal alkene in **8a** furnished highly functionalized 1,3-dienes **20−22** in 62−70% yield[43].

## Preliminary mechanistic studies

To gain some mechanistic insight, we conducted a series of experiments. For the enantioselective cascade coupling pathway, we performed the reaction of 1,3-enyne **1c** with acrylate **2a** in deuterated water (Fig. 6a). The incorporation of deuterium at the stereogenic center indicated that protonation of the alkenyl−Co bond in the metallocycle generated from the first oxidative cyclization occurred (**III→IV**, Fig. 7a). Reaction with deuterated acrylate **2a-D** with $H_2O$ provided product with methyl substituted with one deuterium, suggesting the hydrogen for formation of the methyl group originated from the acrylate and β-H elimination provided the hydrogen (Fig. 6b). Conducting the cascade double alkylation of 1,3-enyne **1d** with acrylate **2a** or **2a-D** in $H_2O$ or $D_2O$ implied that protonation at α position of both esters was involved and none of the C−H bonds in the acrylate was cleaved (Fig. 6c, d). We also surveyed the reason for the formation of a Z/E mixture of the tetrasubstituted alkenes with secondary alkyl-substituted 1,3-enynes (supplementary materials). EPR experiments suggested that an alkenyl radical might be generated through homocleavage of the alkenyl−Co intermediate. Compared with primary alkyl-substituted 1,3-enynes, the more sterically congested secondary alkyl groups might decelerate the rate of

**Fig. 5 | Gram-scale reaction and functionalization. a** The reactions were performed on gram scale. **b, c,** Functionalization of products generated from enantioselective coupling. **d** Functionalization of achiral 1,3-diene. DMA = N,N-dimethylacetamide, DMF = N,N-dimethylformamide, pin = pinacolato, Boc = t-butyloxycarbonyl, m-CPBA = m-chloroperbenzoic acid.

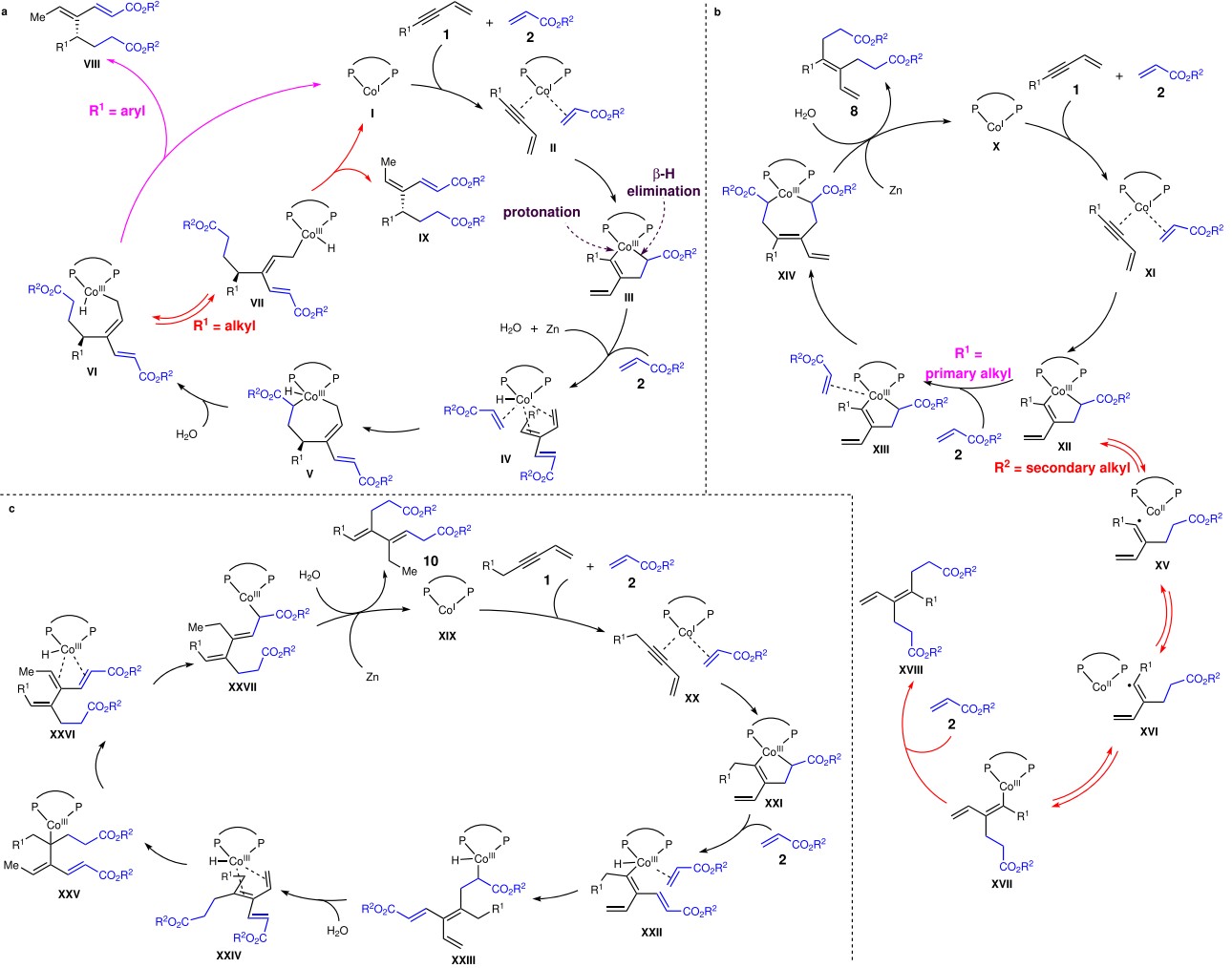

**Fig. 6 | Mechanistic studies. a**, **b** Studies on reactions with aryl-substituted 1,3-enynes. **c**, **d** Studies on reactions to generate products bearing a tetrasubstituted alkene. **e**–**g** Studies on transformations to afford achiral 1,3-dienes containing two trisubstituted olefins. DMA = N,N-dimethylacetamide, DMF = N,N-dimethylforma-mide, NMP = N-methylpyrrolidone.

**Fig. 7 | Proposed catalytic cycles. a** Catalytic cycle for enantioselective cascade coupling and domino 1,6-reduction. **b** Catalytic cycle for double alkylation. **c** Catalytic cycle for cascade coupling to generate 1,3-dienes bearing two trisubstituted alkenes.

insertion of the acrylate to the alkenyl–Co species, leading to competitive homocleavage of the alkenyl–Co(III) bond and subsequent isomerization. For the pathway that generated 1,3-dienes bearing two trisubstituted olefins, we found that similar to the other two pathways, both protonation of α position of esters and transfer of the hydrogen on the acrylate to the terminal alkene to form the methyl group occurred (Fig. 6e, f), whereas the propargyl hydrogen of the 1,3-enyne was rearranged to afford the methylene adjacent to methyl group (Fig. 6g).

Based on all the observations above, we proposed possible catalytic cycles for each pathway (Fig. 7). Co(I) complex underwent regioselective oxidative cyclization to afford metallocycle **III** that was converted to a triene intermediate (**IV**) chelated to the Co center by protonation and β-H elimination (Fig. 7a). A second regio- and enantioselective oxidative cyclization followed by protonation provided allyl–Co(III) species **VI**. With aryl-substituted 1,3-enynes ($R^1$ = aryl), reductive elimination furnished **VIII**, whereas allyl isomerization occurred preferentially to generate **IX** with more sterically congested alkyl-substituted 1,3-enynes ($R^2$ = alkyl). For the pathway of double alkylation, regioselective oxidative cyclization of 1,3-enyne and acrylate to access metallocycle **XII**, which underwent insertion of a second molecule of acrylate followed by protonation to release the product **8** (Fig. 7b). For 1,3-enynes bearing a secondary alkyl group, the larger steric hindrance of the secondary alkyl retarded the insertion of a second molecular acrylate. Homocleavage of the alkenyl–Co(III) bond generated an alkenyl radical intermediate, leading to isomerization of the tetrasubstituted alkene (**XII**→**XV**→**XVI**→**XVII**, Fig. 7b). For the pathway to generate **10**, unlike the pathway shown in Fig. 7b, selective β-H elimination of the metallocycle **XXI** formed from oxidative cyclization occurred in advance of acrylate insertion (Fig. 7c). Protonation of intermediate **XXIII** provided a triene that chelated to the Co center (**XXIV**). Isomerization through Co–H addition and β-H elimination provided a thermodynamically more stable triene bearing two trisubstituted alkenes (**XXIV**→**XXV**→**XXVI**, Fig. 7c). Chemo- and regioselective Co–H addition followed by protonation afforded the product **10**.

In conclusion, a cobalt-catalyzed protocol for cascade coupling of 1,3-enynes with acrylates through divergent pathways to furnish highly functionalized molecules bearing a multisubstituted alkene and/or a stereogenic center in high chemo-, regio- and stereoselectivity has been developed. The transformations proceeded through three distinct reaction modes for cobalt catalysis accurately controlled by ligands. Here we achieve the catalytic pathway-divergent coupling of one molecule of 1,3-enynes with two molecules of electrophilic substituted olefins, reversing the inherent reactivity preference of substrates. The starting materials are inexpensive feedstocks or easily accessible. The tandem reaction pathways enabled the rapid construction of multiple carbon-carbon bonds, editing the carbon skeletons of 1,3-enynes precisely and divergently. The catalysts are derived from inexpensive sustainable cobalt salts and commercially available bisphosphines. The synthetic utility was demonstrated by diverse functionalization of the products, affording a series of useful enantioenriched building blocks that are otherwise difficult to access. Mechanistic studies were conducted to elucidate the reaction mechanisms, revealing that different ligands were able to induce divergent transformations of the metallacycles generated from oxidative cyclization in high chemo-, regio- and stereoselectivity. Such discoveries unveiled unique reaction pathways for cobalt catalysis and domino processes through oxidative cyclization, opening up opportunities for designing cascade reactions promoted by Co-based catalysts, providing a toolbox for complex molecule synthesis, and pushing forward the advancement of organocobalt chemistry. Further investigations on other Co-catalyzed tandem reactions through oxidative cyclization are underway.

## Methods

### General procedure

In a $N_2$-filled glove box, an oven-dried 8-mL vial equipped with a stirring bar was charged with $CoI_2$ (6.3 mg, 0.02 mmol, 10 mol %), **3e** (8.5 mg, 0.02 mmol, 10 mol %), Mn powder (44.0 mg, 0.8 mmol, 4.0 eq) and DMA (0.8 mL). The vial was sealed with a cap (phenolic cap with red PTFE/white silicone septum) and the solution was allowed to stir at room temperature for 30 min. **1a** (49.7 mg, 0.4 mmol, 2.0 equiv.), **2a** (64.9 mg, 0.4 mmol, 1.0 equiv.), $H_2O$ (14.4 mg, 0.8 mmol, 4.0 equiv.) and DMA (0.2 mL) were added to the solution. Then the vial was sealed with a cap (phenolic open-top cap with red PTFE/white silicone septum), and removed from the glove box. The mixture was immediately moved to a thermostatic bath and allowed to stir at 60 °C for 12 h. Upon cooling to room temperature, the reaction mixture was washed with brine (3 × 15 ml), eluted with $Et_2O$ (20 mL), dried over $MgSO_4$, filtered, and concentrated in *vacuo*. The residue was purified by silica gel column chromatography (PE:EA = 40:1) to afford the **4a** as a colorless oil (82.8 mg, 92% yield).

## Data availability

The authors declare that all other data supporting the findings of this study are available within the article and Supplementary Information files, and also are available from the corresponding author on request. Crystallographic data generated during this study have been deposited in the Cambridge Crystallographic Data Center (CCDC) under accession numbers CCDC: 2240339 (**36**), 2250748 (**38**), 2251343 (**40**) and 2240328 (**42**). These data can be obtained free of charge from the CCDC at http://www.ccdc.cam.ac.uk/data_request/cif.

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

## Acknowledgements

This work was financially supported by the National Key R&D Program of China (2022YFA1506100 to F. M.), the National Natural Science Foundation of China (Grant no. 22371294, and 21821002 to F. M.), the Strategic Priority Research Program of the Chinese Academy of Sciences (Grant no. XDB0610000 to F. M.), Beijing National Laboratory for Molecular Sciences (BNLMS202304 to F. M.), the Shanghai Science and Technology Committee (Grant No. 20XD1425000 to Q. C.) and State Key Laboratory of Elemento-Organic Chemistry, Nankai University (Grant no. 202201 to F. M.).

## Author contributions

Conceptualizaton, F.M.; Methodology, F.M., H.W. and X.J.; Investigation, H.W., X.J. and Q.C.; Writing - original draft, F.M.; Writing – review & editing, F.M., Q.C., H.W. and X.J.; Funding Acquisition, F.M. and Q.C.; Supervision, F.M. and Q.C.

## Competing interests

The authors declare no competing interests.
