## [Peer Review File · Nature Communications]

Pathway-divergent coupling of 1,3-enynes with acrylates through cascade cobalt catalysisEditorial Note: This manuscript has been previously reviewed at another journal that is not operating a transparent peer review scheme. This document only contains reviewer comments and rebuttal letters for versions considered at *Nature Communications*.

REVIEWERS' COMMENTS

Reviewer #1 (Remarks to the Author):

I appreciate the efforts the authors have made to revise the manuscript according to the suggestions.

There is only one minor misunderstanding from the previous review, concerning the deuterium incorporation. Even though I appreciate the additional information in the SI, where the authors state that the D-incorporation was determined by comparison to the natural abundance compounds, it still lacks the method that was used (i.e. ^1H NMR, 2D NMR or quantitative ^{13}C NMR). A brief statement in the beginning of the SI is necessary to clarify.

After addition of this information, I can recommend publication.

SHANGHAI INSTITUTE OF ORGANIC CHEMISTRY

Fanke Meng
Professor

March 17, 2024

The following are point-to-point responses to the requests for revision:

For reviewer 1:

“I appreciate the efforts the authors have made to revise the manuscript according to the suggestions. There is only one minor misunderstanding from the previous review, concerning the deuterium incorporation. Even though I appreciate the additional information in the SI, where the authors state that the D-incorporation was determined by comparison to the natural abundance compounds, it still lacks the method that was used (i.e. ^1H NMR, 2D NMR or quantitative ^{13}C NMR). A brief statement in the beginning of the SI is necessary to clarify. After addition of this information, I can recommend publication.”

Our response: Thanks for the comments. We added the information in Page S20 (Supporting Information). They were determined by quantitative ^1H NMR.

Please do not hesitate to contact me if you need any additional information regarding this submission.

Sincerely,
Fanke Meng